# Plasma Vitamin C Levels: Risk Factors for Deficiency and Association with Self-Reported Functional Health in the European Prospective Investigation into Cancer-Norfolk

**DOI:** 10.3390/nu11071552

**Published:** 2019-07-09

**Authors:** Stephen J. McCall, Allan B. Clark, Robert N. Luben, Nicholas J. Wareham, Kay-Tee Khaw, Phyo Kyaw Myint

**Affiliations:** 1National Perinatal Epidemiology Unit, Nuffield Department of Population Health, University of Oxford, Richard Doll Building, Old Road Campus, Oxford OX3 7LF, UK; 2Ageing Clinical & Experimental Research Group, Institute of Applied Health Sciences, University of Aberdeen, Foresterhill, Aberdeen AB25 2ZD, UK; 3Norwich Medical School, University of East Anglia, Norwich NR4 7TJ, UK; 4Department of Public Health and Primary Care, University of Cambridge, Cambridge CB2 0SR UK; 5MRC Epidemiology Unit, University of Cambridge School of Clinical Medicine, Cambridge CB2 0QQ, UK

**Keywords:** vitamin C, self-reported health, risk factors, EPIC-Norfolk

## Abstract

Background: To investigate the demographic and lifestyles factors associated with vitamin C deficiency and to examine the association between plasma vitamin C level and self-reported physical functional health. Methods: A population-based cross-sectional study using the European Prospective Investigation into Cancer-Norfolk study. Plasma vitamin C level < 11 µmol/L indicated vitamin C deficiency. Unconditional logistic regression models assessed the association between vitamin C deficiency and potential risk factors. Associations between quartiles of vitamin C and self-reported functional health measured by the 36-item short-form questionnaire (SF-36) were assessed. Results: After adjustment, vitamin C deficiency was associated with older age, being male, lower physical activity, smoking, more socially deprived area (Townsend index) and a lower educational attainment. Compared to the highest, those in the lowest quartile of vitamin C were more likely to score in the lowest decile of physical function (adjusted odds ratio (aOR): 1.43 (95%CI: 1.21–1.70)), bodily pain (aOR: 1.29 (95% CI: 1.07–1.56)), general health (aOR: 1.4 (95%CI: 1.18–1.66)), and vitality (aOR: 1.23 (95%CI: 1.04–1.45)) SF-36 scores. Conclusions: Simple public health interventions should be aimed at populations with risk factors for vitamin C deficiency. Poor self-reported functional health was associated with lower plasma vitamin C levels, which may reflect symptoms of latent scurvy.

## 1. Introduction

Vitamin C (ascorbic acid) is a co-substrate for many enzymatic reactions. It is essential for the synthesis of collagen proteins and has a vital role in the prevention of bleeding and wound repair [1,2]. Unlike plants and other species, humans cannot synthesise ascorbic acid due to a lack of the functional gulonolactone oxidase enzyme. As a result, it has to be supplemented through the dietary intake of fruit and vegetables [1].

It is important to maintain a sufficient intake of vitamin C to avoid illness. For instance, prolonged deficiency of vitamin C over a 2–3 month period can result in scurvy, a recognised clinical disease [3,4]. The symptoms of scurvy include poor wound repair, haemorrhage, oedema of lower limbs, and fatigue [2]. If the vitamin C deficiency is less severe, latent scurvy can occur without many of the extreme clinical symptoms of scurvy. Latent scurvy presents with more common and non-specific symptoms, such as fatigue, irritability, and muscle pain [5,6]. The non-specificity of the symptoms may cause latent scurvy in the general population to be underreported and underdiagnosed [4,7]. While adult reports of scurvy are very rare in developed nations, 7.1% of a representative sample of the US population were described as having a vitamin C deficiency (<11.4 µmol/L) [7] and other studies define a deficiency as <11 µmol/L [8,9]. Further still, other studies have examined marginal vitamin C deficiency defined as either 11–40 µmol/L [10] or ≥11–28 µmol/L [9,11], where one-fifth of a deprived population, from the United Kingdom, had a suboptimal level of vitamin C [8].

Previous research, in a Western background, has shown that smokers and populations with low socioeconomic status were at increased risk of vitamin C deficiency [8,12]. However, these studies were limited in the number of possible covariates that could be examined for their association with vitamin C deficiency; for example, potential risk factors such as physical activity, educational status, alcohol intake, and prevalent disease were not explored together. The European Prospective Investigation into Cancer (EPIC)-Norfolk cohort study has a wide range of factors that can be explored, which thus offers a more comprehensive exploration of the risk factors of vitamin C.

Symptoms of latent scurvy, such as fatigue, irritability, and muscle pain, are likely to impact self-reported functional health, and if they are caused by vitamin C deficiency, they are easily preventable through supplementation of vitamin C [4,13]. Vitamin C has been shown to be an objective biomarker of fruit and vegetable intake [14]. Low vegetable and fruit intake has been shown to be associated with low self-reported health with regard to summary component scores in the EPIC-Norfolk study [14,15]. However, the association between the level of plasma vitamin C as an objective marker of fruit and vegetable consumption and self-reported health focusing on physical health domains of functional health has yet to be examined.

Therefore, this study has two objectives; firstly, to examine the risk factors for vitamin C deficiency and, secondly, using the 36-item short-form questionnaire (SF-36), to examine the association between plasma vitamin C level and physical functional health in the general population.

## 2. Material and Methods

The study population was drawn from the EPIC-Norfolk cohort study. The study methodology has been previously described [16]. In brief, men and women aged 40–79 were identified from general practices from Norfolk, UK, and were asked to participate in the study by mail. The baseline survey was conducted during 1993–1997 and 30,445 out of 77,630 invited individuals consented to participation. Norwich Local Research Ethics Committee approved the study.

### 2.1. Assay for Vitamin C Measurements

The methods of laboratory analyses and specifically for plasma vitamin C have been previously described [17,18,19]. Non-fasting blood samples were taken from participants at baseline. Venous blood was drawn into plain and citrate bottles and stored overnight in a dark box stored at 4–7 °C. Sample bottles were then centrifuged at 2100*g* at 4 °C for 15 minutes. About 1 year after the initiation of the study, extra blood samples from participants were taken for ascorbic acid assays. Plasma vitamin C was measured from blood taken into citrate bottles, and plasma was stabilized in a standardized volume of metaphosphoric acid and later stored at −70 °C. Plasma vitamin C concentration was estimated ≤1 week after blood sampling using a fluorometric assay [20]. The coefficient of variation was 5.6% at the lower end of the range (mean: 33.2 mol/L) and 4.6% at the upper end (102.3 mol/L). Blood samples for ascorbic acid assays were taken approximately a year into the study.

### 2.2. Baseline Measurements

At the baseline assessment, height and weight were measured using a standardised protocol [21] and these were used to derive baseline BMI. The participants answered a detailed health and lifestyle questionnaire at baseline, including questions on smoking, socioeconomic measures, physical activity, self-reported comorbidities, and a food frequency questionnaire.

Socioeconomic measurements included the Registrar General’s occupation-based classification scheme, educational attainment, and the Townsend index [22]. The Registrar General’s occupation-based classification was reclassified into manual (social classes III manual, IV, and V) and non-manual (or professional) occupations (social classes I, II, III non-manual) [23,24]. Highest educational attainment was included as no attainment, O-level (educational attainment at 16 years), A-level (educational attainment at 18 years), and degree-level or beyond.

The Townsend index is an area level-based deprivation measure. Using the 1991 UK census at enumeration district level, Z-scores were obtained for the following: the percentage of economically active residents aged over 16 years old; percentage of households without a car; percentage of household, not owner-occupied; and percentage of households with more than one person per room were used to calculate the score for enumeration districts. The sum of the Z-scores was used to calculate the Townsend score for each postcode area. The postcodes of participants were used to assign a Townsend score for each participant.

Participants were asked about previous medical conditions using the following question: ‘Has the doctor ever told you that you have any of the following?’. The following prevalent illnesses included in this study were as follows: cancer, stroke, myocardial infarction, diabetes mellitus, and asthma. A food frequency questionnaire was used to derive the alcohol intake [25]. Physical activity was measured using the EPIC short physical activity questionnaire. From this questionnaire, a validated 4-level physical index was created, which was used as a measure of physical activity [26].

### 2.3. Functional Health

The study population was asked to complete the Health and Life Experiences Questionnaire (HLEQ), which included the SF-36 [27]. The questionnaire was sent 18 months after the baseline questionnaire via mail, and the response rate was 73.2% (20,921 participants) of the EPIC-Norfolk sample. Not all participants who attended the baseline health check responded to the SF-36 and vice versa.

The SF-36 contains 36 items, which measures self-reported physical and mental health across a number of domains. These are physical functioning; pain; role limitation due to physical problems; social functioning; role limitation due to emotional problems; mental health; energy/vitality; and general health perception. For the purpose of this study, we chose to examine only those domains related to physical health and vitality. The scoring for each domain is based on the participants’ perceived wellbeing. Each participant is given a score between 0 and 100 for each domain, where a score of 100 represents good health and 0 represents poor health.

### 2.4. Statistical Analysis

Participants with missing data on vitamin C were excluded from the sample. Complete case analysis was used in the analysis. All analyses were completed using STATA 13 SE (College Station, Texas).

#### 2.4.1. Risk Factors for Vitamin C Deficiency

The primary analysis identified risk factors for vitamin C deficiency in the UK setting. This study defined vitamin C deficiency as <11 µmol/L, suboptimal plasma vitamin C levels as ≥11–28 µmol/L, and adequate levels of plasma vitamin C as >28 µmol/L. Unconditional logistic regression models were used, and the dependent variable was vitamin C deficiency (<11 vs. >28 µmol/L) [9]. Log likelihood ratio tests were used to assess linearity in continuous variables. Stepwise methods were used to assess which variables best-predicted vitamin C deficiency. Covariates were chosen for the multivariable model if they had a *p-value* < 0.1 at the univariable level. Covariates were included if they statistically improved the model fit, which was assessed by likelihood ratio tests (<0.05) and if they were associated with the outcome (*p-value* for Wald test < 0.05). Vitamin C supplementation was not assessed as a risk factor for vitamin C deficiency and occupation social class was excluded due to collinearity with educational status (Pearson correlation coefficient > 0.3).

#### 2.4.2. Plasma Vitamin C and Physical Domains of Self-Reported Functional Health

From the 25,639 participants in the baseline question, 18,249 participants also completed HLEQ and, of these, 16,056 had plasma vitamin C samples collected. Histograms were used to assess normality in continuous variables. Descriptive analyses such as chi-square tests (global and test for trend), rank sum tests and Student’s *t*-test were used to assess the difference between participants characteristics, outcomes, and quartiles of vitamin C.

Vitamin C quartiles were the exposure and physical domain scores of SF-36 were the outcomes in this analysis. The physical domains of SF-36 included in this analysis were ‘physical functioning’, ‘role limitation due to a physical problem’, ‘bodily pain’, ‘vitality’ (fatigue), and ‘general health’. These continuous measures of self-reported health were categorised because the distribution was biased to the nearest ten or five. All SF-36 domains were categorised into deciles apart from ‘role limitation due to physical problem’ and this was categorised into quintiles, which was due to the distribution of values.

Multiple logistic regression models were created for each SF-36 physical domain assessed. The physical domains were categorised into a binary variable that compared the bottom decile/quintile to all the other deciles/quintiles. A sensitivity analysis was completed stratifying the fully adjusted model by vitamin C supplementation.

#### 2.4.3. Sensitivity Analyses

In the primary analysis, linear regression models estimated the association between a standard deviation change in vitamin C and each SF-36 domain (continuous). This analysis was further stratified by vitamin C supplementation. A post hoc analysis was completed in participants who did not take vitamin C supplementation; this was completed using spline models. Across the range of vitamin C values, it appeared that the linear association between vitamin C was made up of a number of slopes. As a result, four linear terms were fitted to each SF-36 domain.

## 3. Results

Out of 30,445 participants who agreed to be part of the study, 22,474 attended the second health check and had a plasma vitamin C sample available in the EPIC-Norfolk study.

### 3.1. Risk Factors for Vitamin C Deficiency Analysis

Table 1 shows the sample characteristics according to vitamin C deficiency, and suboptimal and adequate levels of vitamin C. There were 315 (1.4%) participants with a plasma vitamin C < 11 µmol/L and 2410 (10.7%) participants with a plasma vitamin C ≥ 11–28 µmol/L in the EPIC-Norfolk study. At the univariable level, participants with a vitamin C deficiency as compared to those with a normal vitamin C level were older, male, more likely to smoke, be physically inactive, live in a deprived area, have a manual occupation, not take vitamin C supplementation, and have a lower educational status. Other than myocardial infarction, there appeared to be no statistically significant association between vitamin C deficiency and prevalent illness at baseline.

The final regression model included seven risk factors that were statistically associated with vitamin C deficiency; these were older age, being male, physically inactive or current smoking, and having higher Townsend index (higher level of area deprivation), no alcohol intake, or lower educational attainment (see Table 2). Current smokers had a considerably higher risk of vitamin C deficiency than non-smokers (adjusted odds ratio (aOR): 7.38 (95%CI: 5.39–10.12)). Those who were physically inactive had over twice the odds of vitamin C deficiency compared to being physically active (aOR: 2.85 (95%CI: 1.87–4.33)). Similarly, there were twice the odds of vitamin C deficiency in those who did not drink compared to those who had a moderate consumption of alcohol. In addition, after adjusting for confounders, the positive association between excessive drinking and vitamin C deficiency was not statistically significant. Males had over four times the odds of vitamin C deficiency compared to females (aOR: 4.09 (95%CI: 3.11–5.37)). Participants living in the most deprived area had an increased likelihood of vitamin C deficiency compared to those living in the least deprived areas. Furthermore, participants with a lower than O-level educational attainment had over three times the odds of vitamin C deficiency compared to those who had attained a degree (aOR: 3.12 (95%CI: 1.76–5.51)). The adjusted odds for vitamin C deficiency increased by 2% per year increase in age (aOR: 1.02 (95%CI: 1.01–1.04)).

These are the factors remaining in the model after forward stepwise regression method. All factors are mutually adjusted for each other.

### 3.2. Association between Plasma Vitamin C and Self-Reported Physical Health Domains

The characteristics of participants by quartiles of vitamin C are presented in Appendix A. The range of vitamin C in each quartile are as follows: quartile one, <41.0 µmol/L; quartile two, 41.0–53.9 µmol/L; quartile three, 54.0–65.9 µmol/L; and quartile four, ≥66.0 µmol/L. In brief, those in the lowest quartile of vitamin C had very similar characteristics to those with vitamin C deficiency. Table 3 shows the association between quartiles of vitamin C and SF-36 physical functional health domains. In all four models, those in the bottom quartile of vitamin C, compared to the highest, had statistically significant increased odds of having a poor physical functional health SF-36 score. After adjustment for potential confounders, those in the lowest quartile of vitamin C compared the highest quartile had increased odds of having a poor physical functional health score (aOR: 1.43 (95%CI: 1.21–1.70)), a poor bodily pain score (aOR: 1.29 (95%CI: 1.07–1.56)), having poor self-reported general health (aOR:1.4 (95%CI: 1.18–1.66)) and a poor vitality score (aOR:1.23 (95%CI: 1.04–1.45)) (defined as the bottom decile). In addition, after adjustment, those in the lowest quartile of vitamin C had a 1.26 (95%CI: 1.10–1.45) times increased odds of scoring in the lowest quintile of role of physical health score. This trend was more varied when the populations were stratified by those who used vitamin C supplementation (see Table 4). In particular, in those who had a vitamin C supplementation, the results were not statistically significant for two SF-36 physical functional health domains, while the results were more consistent for participants with no vitamin C supplementation.

Appendix A presents the coefficients and 95% confidence intervals of the linear regression models, which treated both vitamin C and SF-36 domains as continuous variables. An increase in one standard deviation (SD) of vitamin C levels was statistically associated with an improved score in all-physical functional health domains assessed (0.65–1.68 unit increase in SF-36 per SD increase in vitamin C) with a larger effect size observed in those who were not taking vitamin C supplementation (1.4–3.5 unit increase in SF-36 per SD increase in vitamin C). Spline models in those without supplementation showed that low vitamin C levels were associated with low SF-36 levels; in particular, the linear association was only present in those below 40 µmol/L.

## 4. Discussion

### 4.1. Main Findings

In this British population-based study, seven factors were associated with vitamin C deficiency. These were older age, being male, lower socioeconomic status measured by higher Townsend index, lower educational attainment, no alcohol intake, smoking, and physical inactivity. This study also showed that lower vitamin C levels were associated with significantly poorer self-reported physical functional health, such as self-reported fatigue, increased bodily pain, and poorer general health. In addition, poorer self-reported health appeared to be worse in those who were not taking vitamin C supplementation and in those who had lower levels plasma of vitamin C at baseline.

### 4.2. Findings in Context—Factors Associated with Vitamin C Deficiency

Consistent with previous research, this study showed that manual occupation as an indicator of low socioeconomic status, smoking, and being male was associated with vitamin C deficiency [8]. In contrast to previous studies [7,8], this study found older age was linearly associated with vitamin C deficiency, where a quadratic trend was shown in the US study [7] and was not examined as a risk factor [8]. In addition to these known risk factors, we also found that physical inactivity increased the risk of vitamin C deficiency. Although no previous research examined this association, this finding is consistent with a wider body of work. Woolcott, et al. (2013) showed that those who were more physically active tended to have a higher fruit and vegetable intake [28].

Conversely, there was no association between excessive alcohol intake and vitamin C deficiency. This is contrast to a previous study which showed an association between excessive alcohol intake and scurvy [29]. A recent review [11] has called for future RCTs to examine vitamin C supplementation in excessive drinkers, which would robustly test this intervention.

Interestingly, in contrast to previous research, BMI was not associated with vitamin C deficiency [7]. This may relate to a relatively narrow range of BMI in this population with very few people in low BMI range (median 25.8 kg/m^2^ (IQR: 23.7–28.3)). Furthermore, this may also be a reflection of inconsistent evidence surrounding the magnitude of the association between BMI and fruit and vegetable intake [15,30].

Our findings showed that no vitamin C supplementation was associated with increased likelihood of a vitamin C deficiency, although this was completed at the univariable level. It should be noted that vitamin C supplementation did not reduce mortality in large-scale RCTs [31,32]; thus, it is probable that vitamin C supplementation may only be beneficial in those with clinical disease as a consequence of a vitamin C deficiency.

### 4.3. Self-Reported Health and Vitamin C

Even after adjustment for sociodemographic factors, lifestyle factors, and comorbidities, an association with low vitamin C quartile and poor functional health remained. This finding was consistent with a wider body of previous work [15] which showed that fruit and vegetable intake was positively associated with improved self-reported functional health using the component summary scores of SF-36. We have further explored this existing knowledge by examining the objective marker of fruit and vegetable intake, vitamin C levels, and individual domains of physical health. The relationship between physical self-reported functional health and vitamin C was most pronounced in low values of plasma vitamin C and in those without supplementation. This is biologically plausible as symptoms of latent scurvy include fatigue, irritability, and muscle pain [4,6,7]. From the outset, those with latent scurvy presenting with bodily pain and fatigue would have low levels of vitamin C, while the association with bodily pain and fatigue would be attenuated at higher levels of vitamin C. Thus, it is likely that supplementation of vitamin C would only have an impact in those with the lowest levels of vitamin C.

### 4.4. Limitations and Strengths

There are a number of limitations to this study. Firstly, the study population had to volunteer to participate, which is shown by a 40% response rate to the baseline questionnaire. As a result, this study may be healthier than the UK population as a whole. However, compared to the general population, the EPIC-Norfolk study was similar to other representative samples other than having a slightly lower proportion of smokers [16]. Furthermore, the external validity of the results may be limited due to the low prevalence of vitamin C deficiency in the EPIC-Norfolk population. Only a single measurement of vitamin C was taken from participants; therefore, the measurements are prone to random error and regression dilution bias. However, random error will only attenuate the relationship between vitamin C and self-reported health. Furthermore, the relationship between vitamin C and self-reported functional health may be biased by residual confounding.

There are a number of limitations regarding the storage of the assays and processing of plasma vitamin C. In particular, recent evidence, which was not known at the time of the study, showed that overnight storage of the sample at 4 °C may result in loss of ascorbate from the sample [33]. As this occurred for all the samples, it will not have caused any differential bias but may likely to have biased estimates towards the null. It could be argued that the vitamin C level in a plasma blood test may fluctuate according to dietary intake and, thus, a fasting blood test may provide a better representation of an individual’s vitamin C level. Furthermore, since this study was completed, there are now more specific methods of plasma vitamin C quantification, such as HPLC, which may improve the accuracy of future studies examining vitamin C [34].

This study has several strengths. We used self-reported physical functional health measures of SF-36, which is a widely validated tool, and we have previously demonstrated that SF-36 is linked to objective health outcomes of mortality, cardiovascular disease, and stroke [35,36,37,38]. In addition, mean age–sex-standardised SF-36 scores from the Health Survey for England and Oxford Healthy Life Survey showed similar values to the EPIC-HLEQ SF-36 scores [39]. Other strengths include a large sample size with the ability to control for a myriad of socioeconomic and lifestyle factors, the ability to examine relationships based on vitamin C supplement use, and use of plasma vitamin C as a measure of dietary vitamin C.

## 5. Conclusions

Low plasma vitamin C levels were associated with poor physical functioning, an established risk factor of objective health outcomes as well as a valid quality of life outcome in its own right. We have also identified risk factors for vitamin C deficiency. Vitamin C deficiency was associated with poor self-reported functional health, which may reflect that this population has symptoms of vitamin C deficiency. Whilst scurvy is rare, populations with poor general health are likely to be micronutrient-deficit and may benefit from increased fruit and vegetable intake.

## Figures and Tables

**Table 1 nutrients-11-01552-t001:** Sociodemographic, lifestyle, and self-reported comorbidities by vitamin C deficiency status.

		Deficiency in Vitamin C Concentration < 11 mol/L; *n* = 315		Suboptimal Vitamin C Concentration ≥ 11–28 mol/L; *n* = 2410		Adequate Vitamin C Concentration > 28 mol/L; *n* = 19,749		Odds Ratio (Deficient vs. Adequate Vitamin C)	95% CI	*p*-Value
Age (mean SD)		62.5	(9.5)	60.7	(9.6)	59.0	(9.2)	1.04	(1.03–1.06)	<0.01
BMI (median IQR)		25.9	(23.4–28.5)	26.4	(24.2–29.1)	25.8	(23.7–28.2)	0.99	(0.96–1.02)	0.57
Number (%)										
Sex	Male	222	(70.5)	1554	(64.5)	8491	(43)	3.16	(2.48–4.04)	<0.01
	Female	93	(29.5)	856	(35.5)	11,258	(57)	1		
Smoking status	Current smoker	120	(38.7)	638	(26.7)	1797	(9.2)	8.82	(6.56–11.86)	<0.01
	Former smoker	118	(38.1)	1045	(43.8)	8298	(42.3)	1.88	(1.40–2.52)	<0.01
	Non-smoker	72	(23.2)	703	(29.5)	9510	(48.5)	1		
Physical activity	Inactive	156	(49.5)	982	(40.7)	5643	(28.6)	3.22	(2.20–4.72)	<0.01
	Moderately inactive	75	(23.8)	581	(24.1)	5764	(29.2)	1.52	(1.00–2.30)	0.05
	Moderately active	52	(16.5)	464	(19.3)	4612	(23.4)	1.31	(0.84–2.05)	0.23
	Active	32	(10.2)	383	(15.9)	3729	(18.9)	1		
Alcohol intake	None	101	(33.3)	568	(24.7)	3594	(18.8)	2.17	(1.66–2.84)	<0.01
	1 to ≤7 units a week	115	(38)	1024	(44.5)	8882	(46.4)	1		
	>7 to ≤14 units a week	41	(13.5)	361	(15.7)	3812	(19.9)	0.83	(0.58–1.19)	0.31
	>14 units a week	46	(15.2)	349	(15.2)	2862	(14.9)	1.24	(0.88–1.75)	0.22
Sociodemographic										
Townsend score	Least deprived (<−3.82)	53	(16.9)	401	(16.7)	3994	(20.3)	1		
	2 (−3.82 to −2.96)	38	(12.1)	421	(17.5)	4034	(20.5)	0.71	(0.47–1.08)	0.11
	3 (−2.96 to − 2.16)	52	(16.6)	472	(19.6)	3922	(19.9)	1.00	(0.68–1.47)	0.99
	4 (−2.16 to −0.74)	62	(19.7)	507	(21.1)	3917	(19.9)	1.19	(0.82–1.73)	0.35
	Most deprived (> −0.74)	109	(34.7)	605	(25.1)	3797	(19.3)	2.16	(1.55–3.01)	<0.01
Education below	Lower than O-level	160	(50.8)	1121	(46.5)	6959	(35.3)	4.48	(2.59–7.75)	<0.01
	O-level	23	(7.3)	224	(9.3)	2050	(10.4)	2.19	(1.12–4.26)	0.02
	A-level or equivalent	118	(37.5)	890	(36.9)	8003	(40.5)	2.87	(1.65–5.01)	<0.01
	Degree or equivalent	14	(4.4)	174	(7.2)	2727	(13.8)	1		
Vitamin C supplementation	No supplementation	242	(77.3)	1824	(76.2)	10,883	(55.4)	2.75	(2.11–3.58)	<0.01
	Taking supplementation	71	(22.7)	570	(23.8)	8772	(44.6)	1		
Social class	Non-manual occupations	112	(37.1)	1116	(47.9)	12,030	(62.1)	1		
	Manual	190	(62.9)	1213	(52.1)	7337	(37.9)	2.78	(2.20–3.52)	<0.01
Self-reported prevalent illness	Cerebrovascular incident	7	(2.2)	61	(2.5)	242	(1.2)	1.83	(0.86–3.91)	0.12
	Myocardial infarction	18	(5.7)	127	(5.3)	551	(2.8)	2.12	(1.31–3.43)	0.002
	Cancer	17	(5.4)	131	(5.4)	1053	(5.3)	1.01	(0.62–1.66)	0.96
	Diabetes mellitus	5	(1.6)	88	(3.7)	411	(2.1)	0.76	(0.31–1.84)	0.54
	Asthma	25	(8)	204	(8.5)	1664	(8.4)	0.94	(0.62–1.42)	0.76

Linear trend between categories. Physical activity shows that there is statistical evidence for a linear trend and no evidence for education level.

**Table 2 nutrients-11-01552-t002:** Risk factors for vitamin C deficiency in EPIC-Norfolk cohort study.

		Adjusted Odds Ratio	95% CI	*p*-Value
**Lifestyle**				
Smoking status	Current smoker	7.38	(5.39–10.12)	<0.01
	Former smoker	1.26	(0.92–1.73)	0.15
	Non-smoker	1		
Physical activity	Inactive	2.85	(1.87–4.33)	<0.01
	Moderately inactive	1.88	(1.20–2.93)	0.01
	Moderately active	1.58	(0.99–2.52)	0.05
	Active	1		
Alcohol intake	None	2.06	(1.55–2.73)	<0.01
	1 to ≤7 units a week	1		
	>7 to ≤14 units a week	0.74	(0.51–1.07)	0.1
	>14 units a week	0.9	(0.62–1.30)	0.57
**Sociodemographic**				
Townsend score	Least deprived (<−3.82)	1		
	2 (−3.82 to −2.96)	0.67	(0.43–1.03)	0.07
	3 (−2.96 to − 2.16)	0.9	(0.60–1.35)	0.62
	4 (−2.16 to −0.74)	1.07	(0.73–1.57)	0.74
	Most deprived (> −0.74)	1.68	(1.18–2.38)	<0.01
Age		1.02	(1.01–1.04)	<0.01
Sex	Male	4.09	(3.11–5.37)	<0.01
	Female	1		
Education	Lower than O level	3.12	(1.76–5.51)	<0.01
	O-level	1.81	(0.90–3.63)	0.1
	A-level or equivalent	2.33	(1.32–4.11)	<0.01
	Degree or equivalent	1		

These are the factors remaining in the model after forward stepwise regression method. All factors are mutually adjusted for each other.

**Table 3 nutrients-11-01552-t003:** Association between plasma vitamin C quartile and SF-36 domain (physical functions and vitality).

	Vitamin Quartile	Unadjusted			Model A			Model B			Model C			Model D		
		Odds ratio	95% CI	*p*-value	Odds ratio	95% CI	*p*-value	Odds ratio	95% CI	*p*-value	Odds ratio	95% CI	*p*-value	Odds ratio	95% CI	*p*-value
Physical function*	1 (Lowest)	2.03	(1.75-2.35)	<0.01	2.17	(1.85–2.54)	<0.01	1.75	(1.48–2.06)	<0.01	1.56	(1.32–1.84)	<0.01	1.43	(1.21–1.70)	<0.01
	2	1.28	(1.10–1.51)	<0.01	1.42	(1.21–1.68)	<0.01	1.3	(1.09–1.54)	<0.01	1.16	(0.97–1.38)	0.1	1.11	(0.93–1.32)	0.24
	3	0.91	(0.77–1.07)	0.24	0.99	(0.84–1.17)	0.92	0.96	(0.80–1.14)	0.61	0.89	(0.75–1.06)	0.2	0.88	(0.73–1.05)	0.15
	4 (Highest)	1			1			1			1			1		
Role physical health**	1 (Lowest)	1.6	(1.41–1.81)	<0.01	1.63	(1.43–1.87)	<0.01	1.41	(1.23–1.62)	<0.01	1.33	(1.16–1.53)	<0.01	1.26	(1.10–1.45)	0.001
	2	1.16	(1.02–1.32)	0.02	1.23	(1.07–1.41)	<0.01	1.15	(1.00–1.32)	0.05	1.09	(0.95–1.25)	0.23	1.06	(0.92–1.22)	0.42
	3	0.98	(0.86–1.12)	0.74	1.04	(0.91–1.18)	0.60	1	(0.88–1.15)	0.95	0.98	(0.85–1.12)	0.72	0.97	(0.84–1.11)	0.62
	4 (Highest)	1			1			1			1			1		
Bodily Pain*	1 (Lowest)	1.68	(1.42–2.00)	<0.01	1.83	(1.53–2.18)	<0.01	1.49	(1.24–1.79)	0.001	1.37	(1.13–1.65)	<0.01	1.29	(1.07–1.56)	0.01
	2	1.19	(0.99–1.42)	0.06	1.29	(1.08–1.55)	0.01	1.19	(0.99–1.44)	0.07	1.1	(0.91–1.33)	0.34	1.07	(0.89–1.30)	0.47
	3	1	(0.84–1.20)	0.98	1.06	(0.88–1.27)	0.54	0.99	(0.82–1.20)	0.94	0.95	(0.79–1.15)	0.59	0.95	(0.78–1.14)	0.57
	4 (Highest)	1			1			1			1			1		
Vitality*	1 (Lowest)	1.35	(1.17–1.57)	<0.01	1.6	(1.38–1.87)	<0.01	1.34	(1.14–1.58)	<0.01	1.29	(1.10–1.52)	<0.01	1.23	(1.04–1.45)	0.01
	2	1.12	(0.96–1.30)	0.15	1.26	(1.08–1.47)	<0.01	1.16	(0.99–1.36)	0.07	1.12	(0.95–1.32)	0.17	1.1	(0.94–1.30)	0.23
	3	0.98	(0.85–1.15)	0.84	1.04	(0.90–1.22)	0.59	1.03	(0.88–1.20)	0.75	1.01	(0.86–1.18)	0.92	1	(0.85–1.17)	0.96
	4 (Highest)	1			1			1			1			1		
General health*	1 (Lowest)	2.01	(1.72–2.35)	<0.01	1.92	(1.64–2.26)	<0.01	1.56	(1.32–1.84)	<0.01	1.48	(1.25–1.75)	<0.01	1.4	(1.18–1.66)	<0.01
	2	1.38	(1.17–1.62)	<0.01	1.36	(1.15–1.60)	<0.01	1.23	(1.04–1.46)	0.02	1.17	(0.99–1.39)	0.07	1.14	(0.96–1.36)	0.13
	3	1	(0.84–1.19)	0.99	1	(0.84–1.19)	0.99	0.95	(0.79–1.13)	0.54	0.92	(0.77–1.10)	0.37	0.91	(0.76–1.09)	0.31
	4 (Highest)	1			1			1			1			1		

* Outcome defined as bottom decile vs. all other deciles; ** Outcome defined as the bottom quintile vs. rest; Model A: Basic model + age and sex. Model B: model B + smoking, physical activity and alcohol intake. Model C: model B + BMI. Model D: model C + education status, Townsend index and previous medical history (cancer, diabetes, cerebrovascular incident, myocardial infarction, asthma).

**Table 4 nutrients-11-01552-t004:** Association between plasma vitamin C quartile and SF-36 domain (physical function and vitality) stratified by supplementation of vitamin C.

	Vitamin Quartile	With Vitamin C Supplementation			No Vitamin C Supplementation		
		Odds ratio	95% CI	*p*-Value	Odds ratio	95% CI	*p*-Value
Physical function	1 (Lowest)	1.62	(1.26–2.08)	<0.01	1.45	(1.12–1.86)	<0.01
	2	1.05	(0.82–1.35)	0.68	1.26	(0.97–1.64)	0.08
	3	0.93	(0.74–1.19)	0.58	0.88	(0.66–1.16)	0.36
	4 (Highest)	1			1		
Role physical health	1 (Lowest)	1.19	(0.96–1.48)	0.11	1.47	(1.20–1.81)	<0.01
	2	0.97	(0.80–1.19)	0.8	1.24	(1.01–1.54)	0.04
	3	0.98	(0.82–1.18)	0.85	1.01	(0.81–1.25)	0.94
	4 (Highest)	1			1		
Bodily Pain	1 (Lowest)	1.36	(1.02–1.81)	0.03	1.32	(1.00–1.73)	0.05
	2	1.15	(0.88–1.51)	0.3	1.08	(0.81–1.43)	0.6
	3	1.07	(0.83–1.38)	0.59	0.84	(0.63–1.14)	0.27
	4 (Highest)	1			1		
Vitality	1 (Lowest)	1.2	(0.92–1.56)	0.17	1.34	(1.06–1.69)	0.01
	2	0.98	(0.77–1.25)	0.9	1.28	(1.01–1.61)	0.04
	3	0.99	(0.80–1.22)	0.9	1.04	(0.82–1.33)	0.72
	4 (Highest)	1			1		
General health	1 (Lowest)	1.52	(1.16–2.00)	<0.01	1.34	(1.05–1.71)	0.02
	2	1.05	(0.81–1.37)	0.71	1.21	(0.94–1.55)	0.14
	3	1.03	(0.81–1.31)	0.8	0.81	(0.62–1.05)	0.12
	4 (Highest)	1			1		

Adjusted for age, sex, smoking, physical activity, alcohol intake, BMI, education status, Townsend index, and previous medical history (cancer, diabetes, cerebrovascular incident, myocardial infarction, asthma)). All outcomes compared the bottom decile vs. all other deciles except role physical health, which was defined the comparison as the bottom quintile vs. rest.

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
