# Peer review of "Plasma Vitamin C Levels: Risk Factors for Deficiency and Association with Self-Reported Functional Health in the European Prospective Investigation into Cancer-Norfolk"

_nutrients, 2019, doi:10.3390/nu11071552_

Reviewer 1 Report

In an impressive large cross-sectional UK population study (n=22,740) the authors have interrogated plasma vit  C levels and a self-reported detailed health and lifestyle questionnaire including smoking, socioeconomic status, age, exercise-activity, food-frequency and comorbidity items in order to identify risk factors for vit C deficiency and associations of vit C levels with physical functional health. The authors were able to identify 7 factors related to vit C deficiency (399 subjects were deficient compared to 2,075 that were sufficient). In addition, poorer self-reported health appeared to be worse in those subjects not taking vit C supplements and those subjects with lower vit  C levels. 

There is now a re-awakening of interest in vit C  (e.g., ICU, cancer therapy, interactions with other “anti-oxidants”,  tissue culture challenges, etc).  This paper . should be a valuable literature addition,  especially as reliable vitamin C determinations  presents challenges and are not frequently determined by hospital clinical labs.

                                                                 Comments

 1. The authors quote a vitamin C plasma level 11.4 µ/L as being a “deficiency level” (ref. 4). Are there other recommended levels of note in the literature that are worth mention? Below 11.4 would seem to define a rather severe deficiency. 

2. Range of vitamin C level determinations in the 4 quartiles should clearly be given in text of Results and even in Discussion in paragraph that discusses more than just the vit C  level for severe deficiency. Surely there are recommendations for more health levels of vitamin C out there. 

3. Methods of determinations of vitamin C need to be more detailed in a  paper focused on vitamin C determinations.

4.  What is “absolute” vs “non-absolute” level of vitamin C (line 210, page 5).

Author Response

1.       The Authors’s quote a vitamin C plasma level 11.4 µ/L as being a “deficiency level” (ref. 4). Are there other recommended levels of note in the literature that are worth mention? Below 11.4 would seem to define a rather severe deficiency.

Authors’ response:  We thank the reviewer for their comments and positive review. As with all cut-offs, they are used as a threshold to determine whether a person requires treatment in a clinical setting or they are used to help improve the interpretability of the findings in the research context.  Many papers have used vitamin C as a continuous variable, which we have also completed in the supplementary section (Tracica et al. 2017). Richardson (2002) provided a range of scurvy range “<11 µmol/l”, normal range “40-100 µmol/l” and hypovitaminosis “11-40 µmol/l”. Others have suggested having a marginal vitamin C deficiency on serum levels of >11 – 28 µmol/l  (Lim et al. 2018). As a result, we have examined this intermediate group in the analysis. 

We have added the following text to the introduction.

Other studies have examined marginal vitamin C deficiency defined as either 11-40 µmol/L  [5] or >11-28 µmol/L [6].”

2.       Range of vitamin C level determinations in the 4 quartiles should clearly be given in the text of Results and even in Discussion in the paragraph that discusses more than just the vit C  level for severe deficiency. Surely there are recommendations for more health levels of vitamin C out there.

Authors’ response: We have added the following text to the results section:

The range of vitamin C in each quartile are as follows quartile one:<41.0 µmol/L; quartile two: 41.0 -53.9 µmol/L; quartile three: 54.0 – 65.9 µmol/L; and quartile four: ≥66.0 µmol/L.”

3. Methods of determinations of vitamin C need to be more detailed in a  paper focused on vitamin C determinations.

Authors’ response:  We thank the reviewer for this comment. These have been added to the methods:

Non-fasting blood samples were taken from participants at baseline. Venous blood was drawn into plain and citrate bottles and they were stored overnight in a dark box at 4°C - 7°C, sample bottles were then centrifuged at 2100 g at 4°C for 15 minutes. About 1 year after the initiation of the study, extra blood samples from participants were taken for ascorbic acid assays. Plasma vitamin C was measured from blood taken into citrate bottles, and plasma was stabilized in a standardized volume of metaphosphoric acid and later stored at -70°C. Plasma vitamin C concentration was estimated ≤1 week of blood sampling using a fluorometric assay [15]. The coefficient of variation was 5.6% at the lower end of the range (mean: 33.2 mol/L) and 4.6% at the upper end (102.3 mol/L).

4.  What is “absolute” vs “non-absolute” level of vitamin C (line 210, page 5)

Authors’ response: Thank you for this comment we have removed the word absolute from the discussion.

References

Richardson, T.; Ball, L.; Rosenfeld, T. Will an orange a day keep the doctor away? Postgrad. Med. J. 2002, 78, 292–294

Travica, Nikolaj, et al. "Vitamin C status and cognitive function: A systematic review." Nutrients9.9 (2017): 960.

Lim, Daniel James, Yogesh Sharma, and Campbell Henry Thompson. "Vitamin C and alcohol: a call to action." BMJ Nutrition, Prevention & Health (2018): bmjnph-2018.

Reviewer 2 Report

This paper describes an analysis of the EPIC-Norfolk cohort, focussed on a single point analysis of plasma vitamin C levels and self-reported health status. Given the large number of individuals sampled and the strength of the overall background of the EPIC-Norfolk study, this is an interesting study and considerable effort has been put into the analysis. Overall, the results are not surprising: low vitamin C status is associated with older age, lower income levels, and higher alcohol intake. In addition, low vitamin C status was associated with increased risk of poor physical function, pain, decreased general health and lower vitality scores.

Although interesting, the study has a number of major deficiencies that relate largely to the measurement of vitamin C. Firstly, the method used is not described. It would not be difficult to add a sentence to indicate what method was used. Also, the timing of the measurement is not indicated and it is difficult to determine this from the information given – I assume this was carried out at first recruitment (1993-97) and referred to in the 1999 paper, but this is unclear.

More troublesome is the potential for inaccuracies in the vitamin C levels measured. As indicated in the supplementary information, the blood samples were non-fasting, and vitamin C levels are therefore likely to be confounded by the most recent food intake. That this affects the results is clear from the range of values given in Supplementary Table 1. The highest quartile has levels ranging from 66-242 µM, which is well above the 100 µM saturation level, and indicates that many samples will be influenced by recent dietary intake.

In addition, there is a distinct possibility that processing of the blood samples introduced further errors in vitamin C measurement. The blood samples were stored at 4oC overnight before processing (see supplementary methods). As pointed out in a recent publication (Pullar et al, Antioxidants 2018, 7, 29; doi:10.3390/antiox7020029), overnight storage even at 4oC, could result in significant loss of ascorbate from the sample.

These issues above require specific and extensive discussion to take account of the uncertainty that results for interpretation. The short paragraph in the Discussion (p11, lines 271-275) is not sufficient.

Descriptions of vitamin C status can be confusing in the literature. For this paper, which is based on an interpretation of the effects of variations in vitamin C status, it would be helpful to more clearly define what is meant by deficiency. In the introduction, the terms latent scurvy and scurvy are loosely used. These could be more tightly defined, with clarification as to the distinction between scurvy as a clinical disease due to deficiency, and latent scurvy which may reflect low status without many of the extreme clinical symptoms.

All points above need to be carefully and fully addressed.

Author Response

1.       Although interesting, the study has a number of major deficiencies that relate largely to the measurement of vitamin C. Firstly, the method used is not described. It would not be difficult to add a sentence to indicate what method was used.

Authors’ response:  We have added a paragraph to the methods section to provide more detail on the measurement of vitamin C. Please also refer to our response to the reviewer one. 

2.       Also, the timing of the measurement is not indicated and it is difficult to determine this from the information given – I assume this was carried out at first recruitment (1993-97) and referred to in the 1999 paper, but this is unclear.

Authors’ response: Yes the samples were taken at the baseline.

We have added the following sentence to the methods section:

Blood samples for ascorbic acid assays were taken approximately a year into the study.”

3.       More troublesome is the potential for inaccuracies in the vitamin C levels measured. As indicated in the supplementary information, the blood samples were non-fasting, and vitamin C levels are therefore likely to be confounded by the most recent food intake. That this affects the results is clear from the range of values given in Supplementary Table 1. The highest quartile has levels ranging from 66-242 µM, which is well above the 100 µM saturation level, and indicates that many samples will be influenced by recent dietary intake.

Authors’ response: We agree that this is a limitation of the study; however, only 1.5% of the sample were above 100 µM and 0.09% were above 150 µM. Measurement error occurs in ever epidemiological study adding noise to the effect estimates; however, this error is not systematic and will reflect habitual consumption. It is reassuring that our results are consistent with previous studies examining the risk factors of vitamin C deficiency.

Is recent dietary intake of vitamin C not a good predictor of previous dietary intake of vitamin C? We would argue that recent dietary intake would be collinear with historical dietary intake rather than a confounder. Yes, there may be some misclassification and we appreciate that this is a limitation of the analysis and misclassification, like all epidemiological studies, may have occurred across our exposure group. In order to account for this possible misclassification, we have restricted the reference group to those who had a plasma vitamin C  ≥28 µmol/L.

We have added more comments to the limitation.

In particular, recent evidence, which was not known at the time of the study, showed that overnight storage of the sample at 4 degrees Celsius may result in loss of ascorbate from the sample [35]. As this occurred for all the sample it is will not have caused any differential bias but may likely to have biased estimates towards the null. It could be argued that the vitamin C level in a plasma blood test may fluctuate by dietary intake, thus a fasting blood test may provide a better representation of an individual’s vitamin C level. Furthermore, since this study was completed there are more specific methods of plasma vitamin C quantification such as HPLC, which may improve the accuracy of the results of future studies examining vitamin C [36].”

4.       In addition, there is a distinct possibility that processing of the blood samples introduced further errors in vitamin C measurement. The blood samples were stored at 4oC overnight before processing (see supplementary methods). As pointed out in a recent publication (Pullar et al, Antioxidants 2018, 7, 29; doi:10.3390/antiox7020029), overnight storage even at 4oC, could result in significant loss of ascorbate from the sample.

These issues above require specific and extensive discussion to take account of the uncertainty that results for interpretation. The short paragraph in the Discussion (p11, lines 271-275) is not sufficient.

Authors’ response: We thank the reviewer for this important point and we have added the following information to the limitations section. We have added the following information to the limitations section:

“In particular, recent evidence, which was not known at the time of the study, showed that overnight storage of the sample at 4 degrees Celsius may result in loss of ascorbate from the sample [27]. As this occurred for all the sample it is will not have caused any differential bias but may likely to have biased estimates towards the null.”

5.       Descriptions of vitamin C status can be confusing in the literature. For this paper, which is based on an interpretation of the effects of variations in vitamin C status, it would be helpful to more clearly define what is meant by deficiency.

Author’s response: We agree that the literature has many cutoffs for this continuous phenotype. We thank the reviewer for this comment we have added these sentences to the method.

This study defined vitamin C deficiency as<11 sub-optimal="" plasma="" vitamin="" c="" levels="" as="" 11-28="" l="" and="" adequate="" of="">28 µmol/L.”

6.       In the introduction, the terms latent scurvy and scurvy are loosely used. These could be more tightly defined, with clarification as to the distinction between scurvy as a clinical disease due to deficiency, and latent scurvy which may reflect low status without many of the extreme clinical symptoms.

Author’s response: We thank the reviewer for this comment we have added this paragraph to the introduction to address this point:

“Scurvy is a rare clinical disease [3], which is the result of a deficiency in vitamin C intake over a 2-3 month period [4]. The symptoms of scurvy include poor wound repair, haemorrhage, oedema of lower limbs and fatigue [5]. While latent scurvy reflects a less severe vitamin C deficiency without many of the extreme clinical symptoms. Latent scurvy presents with more common and non-specific symptoms such as fatigue, irritability and muscle pain  [6,7], which may cause latent scurvy in the general population to be underreported and under-diagnosed [8].”

All points above need to be carefully and fully addressed.

We thank the reviewer for their comments, which have improved the content of the paper. We also hope the above responses are satisfactory to the reviewer.

Reviewer 3 Report

Overall I enjoyed reading this manuscript, and feel that it brings needed attention to the idea that micronutrient intake is an important determinant of health. Much emphasis is placed on caloric intake, fat vs. carbs, and other macronutrient-related concerns, but studies like this one are important for reminding us that cutting calories and fat isn't perhaps as important as making sure that we include some micronutrients in those calories and fat.

I do feel that the main message here is that micronutrients are important. I wouldn't say that you could reasonably use the results of this study to identify people at-risk for vitamin C deficiency, but if you change the focus of this paper to emphasize that micronutrient deficiency is a risk factor for reduced functional status I'd be totally on board. :o)

Abstract: 

Sentence in lines 21 and 22 is a fragment and needs to be re-worded

Line 21: Stating that the aim is to investigate “the factors” associated with vitamin C deficiency seems vague - might help to clarify that these are demographic and lifestyle factors 

Within the abstract, the conclusions don’t match the results. The results here don’t say anything about using risk factors to identify individuals with vitamin C deficiency. The results are based on vitamin C deficiency that was determined using a blood test.

The conclusions in this abstract really don’t match the results of the paper themselves; you didn’t in fact identify at-risk people; you identified risk factors for vitamin C deficiency. 

In the abstract, it sounds like latent scurvy was diagnosed using the SF-36. This is not a valid method to determine whether a person has latent scurvy; low SF-36 scores can be related to all sorts of problems, so saying they’re due to scurvy seems a stretch. 

Introduction: 

Line 42: Humans are animals… :o) 

Line 48: The SE for the prevalence of vitamin C deficiency isn’t especially informative here - a confidence interval is easier to interpret than an SE when describing the precision of an estimate. Although I’m not really used to seeing either a CI or an SE in this sort of context. 

Line 50: What do you mean by “deprived populations”? 

Lines 50 to 52: These last sentences in this paragraph don’t really clarify the research gap. More detail is needed to clarify what those previous studies did and didn’t do. 

Line 53: Define latent scurvy 

Line 54: Start second sentence by saying “Latent scurvy is likely to…” - if you refer only to the symptoms it sounds like you’re saying that such symptoms will always be addressed by fruit/vegetable consumption, even if they’re not related latent scurvy. 

Overall my feeling is that this introduction needs to be re-structured - the ideas jump around rather than build logically. 

In terms of the goals of this paper (lines 62-64), the first goal is reasonable, but the second could perhaps use a qualifier. This cross-sectional analysis cannot determine temporality or causality, and it seems quite likely that vitamin C deficiency is a marker for many other risk factors (e.g., low SES, low literacy, poor health care access, poor diet quality, etc.) that may impact physical functional health.

Materials and Methods: 

References should be included for the socioeconomic measures used in this analysis, if available (e.g., the Townsend index). 

Line 100: Replace “this” with “which”

Lines 102-103: What do you mean by “vice versa” - why were there participants who completed the SF-36 but didn’t do the baseline health check? Wasn’t being in the baseline study necessary to be invited to do the follow-up survey? 

Line 108: Why only focus on physical functioning and vitality? For example, irritability is a symptom of latent scurvy, and would fall under the “role limitation due to emotional problems” domain. 

Line 116: This wouldn’t be a case-control analysis, as this is a cross-sectional study. Cross-sectional studies look at prevalent outcomes (individuals with vitamin C deficiency regardless of when it developed) while a case-control analysis would include only incident cases. You don’t know how long individuals in this population had vitamin C deficiency, so you wouldn’t call this a case-control analysis. 

Line 118: These two vitamin C categories (<11.4 vs. ≥11.4 μmol/L) likely include some misclassification around the cutpoint. You could consider looking at low (<11.4) vs.="" optimal="">22.7). Removing the middle group (considered “marginal”) will arguably give you more meaningful comparison groups. 

Line 118: What do you mean by “assess linearity in categorial variables”? 

Line 119: Better to say “Stepwise methods were used to assess which variables…”

Line 122: Don’t need to specify that covariates were added sequentially to the model, as you already explain that you used a stepwise model. 

Line 123: replace “there” with “they” 

Overall thought about model building process: I’ve used stepwise selection myself, but more and more I’m thinking of it as a part of the process rather than as a way to identify the final model. Giving the computer free rein necessarily means not using your own knowledge to identify which variable make most sense in the model. For example, following is a link that details some concerns about stepwise models and suggests some alternatives: https://towardsdatascience.com/stopping-stepwise-why-stepwise-selection-is-bad-and-what-you-should-use-instead-90818b3f52df

Lines 127-129: I understand what you’re saying here, but had to read a few times - try re-wording this for better clarity. 

Lines 136-146: Why are you comparing the bottom decile/quartile to all other categories combined? My feeling is that it would be more appropriate to use one of the extreme categories as your reference group, and to show the ORs for all other categories relative to that reference group. This would allow you to look for trends and threshold effects.

Results: 

Line 159: Consider including a third group here - those with optimum vitamin C levels (>22.7)

Line 165: Referring to a baseline time point here may confuse some readers - this is a cross-sectional analysis that happens to have measured different variables at two different time points. There wasn’t really a follow-up time point, just a second assessment time point. My feeling is that it would be better to simple remove reference to “baseline” here, as it may confuse people about the nature of this study design. 

Line 166: It doesn’t seem appropriate to say “after adjustment” here, because you don’t have one key independent variable of interest. You might say “The final regression model included seven risk factors…” 

Lines 166-180: These results are re-stating the information in Table 2. This can be shortened into a brief summary of these results. 

Line 183: Include the word “to” before “those with”. 

Lines 182-184: Why is this supplemental table necessary? It should be referred to with Table 1 if it is kept, but I’d argue against keeping it as it seems redundant. 

Lines 184-192: Don’t re-state results presented in Table 3 - just summarize 

Line 198: How much did these domains improve? Don’t need to re-state all these results, but could provide the range of improvement across the different domains. 

Table 3: Consider presenting the three categories of vitamin C status (deficiency, borderline, optimum) rather than quartiles; if quartiles are kept, include the ranges for each quartile (i.e., include the range for vitamin C levels within each quartile).

Table 3: Model A should be described as being adjusted for age and sex, not “unadjusted + age and sex” (it is adjusted, so calling it unadjusted seems confusing) 

Table 4: Given that all models are adjusted for the same covariates, you could label the first column “independent variable” and include an asterisk there that refers to the covariates included in the models.

Discussion: 

Line 207: Delete “those with”

Line 210: Delete the first “who”  

Line 215: Say “in contrast to previous studies” then included the citations (although the way you did it is rather fun - it just took me a second to get it)

Line 215: What exactly did those previous studies find? A non-linear association with age? 

Line 217: Have previous studies looked for this, or have there simply not been any studies of this association? 

Line 218: Delete “has” 

Lines 232-233: How many people in this study would really qualify as excessive drinkers? If 14 units per week refers to having two standard drinks a day, then that’s not excessive for men. It would be interesting to categorize drinking more formally, with separate designations for men and women. For example, the CDC defines heavy drinking as 8 or more drinks per week for women, and 15 or more per week for men. Binge drinking could also be considered (if such data are available). 

Lines 236-238 do not seem to fit with the rest of this paragraph 

Line 245: Include a comma after comorbidities 

Lins 246-247: It seems contradictory to say that a finding is both novel and consistent with previous work. 

Line 247: Definitely too confusing to start a sentence with a parenthetical citation

Lines 253-254: Why are all these authors listed here?  

Lines 257-258: Not sure what you mean here. Is “phonotypical” the correct word? The sentence makes sense without that word, but with it I’m not sure what’s going on. :o)

Lines 260-270: This paragraph jumps around a bit - would help to re-structure so that the information is presented more logically.

Lines 267-270: One method to address this would be to compare the highs vs. the lows, as I suggested previously. 

Lines 271-275: While you’re re-structuring, include the information about misclassification of vitamin C (lines 267-270) in this paragraph.  

Line 277: What does “it” refer to? The SF-36 data? 

Line 280: I’m not sure that the “objective” part of the plasma vitamin C measure is really the key strength. You described some important limitations of this measure, but is there any information on how valid these data are likely to be? In line 272 you say “it could be argued that vitamin C levels may fluctuate” - is there any evidence of that? I’m more interested in knowing just how good this biomarker is than in the fact that it’s objective. There are some truly terrible objective biomarkers of nutrition status out there. How good is this one?

Conclusions: 

Line 282: This analysis was about vitamin C levels - you didn’t look at whether vitamin C levels were in fact an indicator of fruit and vegetable intake in this population. Which perhaps is fine, as that’s been shown previously, but in this context it seems clearer to simply say “Low plasma vitamin C levels were associated with…” 

Overall these conclusions seem reasonable, but I encourage a little more consideration of the fact that the risk factors you identified here are very broad and will relate to many adverse health outcomes. I don’t really feel that you can pinpoint vitamin C deficiency with what you showed here. If a patient comes in complaining of poor health, and the patient is an older male who smokes and drinks and is inactive, it would be odd to say “aha! must be that you have low vitamin C!” - it could just be drinking and smoking and lack of activity. 

I feel like the better approach to identifying those at risk for low vitamin C is simply to ask whether people eat fresh fruits and vegetables. If they don't, then worry about their vitamin C. 

My biggest concern here is that you frame these results in the context of a public health intervention to identify those at risk for vitamin C deficiency. I don't feel that these results will be useful in that sense (the identified risk factors are not specific enough), but I do feel that this would be a strong paper if the emphasis was shifted to a discussion of how vitamin C deficiency is associated with poor functional health in general. Even if people don't have scurvy, they are going to have general poor health if they're micronutrient deficient. 

Author Response

Reviewer three

Overall I enjoyed reading this manuscript, and feel that it brings needed attention to the idea that micronutrient intake is an important determinant of health. Much emphasis is placed on caloric intake, fat vs. carbs, and other macronutrient-related concerns, but studies like this one are important for reminding us that cutting calories and fat isn't perhaps as important as making sure that we include some micronutrients in those calories and fat.

I do feel that the main message here is that micronutrients are important. I wouldn't say that you could reasonably use the results of this study to identify people at-risk for vitamin C deficiency, but if you change the focus of this paper to emphasize that micronutrient deficiency is a risk factor for reduced functional status I'd be totally on board. :o)

We thank the reviewer for their positive comments about the paper. Their comments have made this study more readable and robust so we thank them for their time.

Abstract:

Sentence in lines 21 and 22 is a fragment and needs to be re-worded

Authors’ response: We have rephrased to:

“To investigate the factors associated with vitamin C deficiency and to examine the association between plasma vitamin C level and self-reported physical functional health.”

Line 21: Stating that the aim is to investigate “the factors” associated with vitamin C deficiency seems vague - might help to clarify that these are demographic and lifestyle factors

Authors’ response: We have rephrased to:

“To investigate the demographic and lifestyles factors associated with vitamin C deficiency and to examine the association between plasma vitamin C level and self-reported physical health.”

Within the abstract, the conclusions don’t match the results. The results here don’t say anything about using risk factors to identify individuals with vitamin C deficiency. The results are based on vitamin C deficiency that was determined using a blood test.

The conclusions in this abstract really don’t match the results of the paper themselves; you didn’t in fact identify at-risk people; you identified risk factors for vitamin C deficiency.

Authors’ response: We have rephrased to: “Simple public health interventions should be aimed at populations with risk factors for vitamin C deficiency.”

In the abstract, it sounds like latent scurvy was diagnosed using the SF-36. This is not a valid method to determine whether a person has latent scurvy; low SF-36 scores can be related to all sorts of problems, so saying they’re due to scurvy seems a stretch. 

Authors’ response: We have changed the tone of the conclusion:

“Apparent poor self-reported functional health was associated with lower plasma vitamin C levels, which may reflect symptoms of latent scurvy.”

Introduction:

Line 42: Humans are animals… :o)

Authors’ response: we have rephrased to:

“Unlike plants and other species”

Line 48: The SE for the prevalence of vitamin C deficiency isn’t especially informative here - a confidence interval is easier to interpret than an SE when describing the precision of an estimate. Although I’m not really used to seeing either a CI or an SE in this sort of context.

Authors’ response: we have removed the SE.

Line 50: What do you mean by “deprived populations”?

Authors’ response: we have rephrased to:

“Previous research has shown that smokers and those with low socioeconomic status”

Lines 50 to 52: These last sentences in this paragraph don’t really clarify the research gap. More detail is needed to clarify what those previous studies did and didn’t do.

Authors’ response: We thank the reviewer for the comment. We have added the following sentences to the introduction:

“However, these studies were limited in the number of possible covariates that could be examined for their association with vitamin C deficiency, for example, potential risk factors such as physical activity, educational status, alcohol intake and prevalent disease were not explored together. The EPIC-Norfolk cohort study has a wide range of factors that can be explored which provides a more comprehensive exploration of the risk factors of vitamin C.”

Line 53: Define latent scurvy

Author’s response: Thank you for the comment we have added this paragraph.

“Scurvy is a rare clinical disease that results in anaemia, internal bleeding fatigue and gum disease [8]. While latent scurvy reflects a vitamin C deficiency without many of the extreme clinical symptoms.  Latent scurvy presents with more common and non-specific symptoms such as fatigue, irritability and muscle pain  [9,10], which may cause latent scurvy in the general population to be underreported and under-diagnosed [4,11].”

Line 54: Start second sentence by saying “Latent scurvy is likely to…” - if you refer only to the symptoms it sounds like you’re saying that such symptoms will always be addressed by fruit/vegetable consumption, even if they’re not related latent scurvy.

Authors’ response: We agree and made a clarification in the sentence.

“Symptoms of latent scurvy, such as fatigue, irritability and muscle pain, are likely to impact self-reported functional health and if they are caused by vitamin C deficiency are easily preventable through supplementation of vitamin C [4,8].”

Overall my feeling is that this introduction needs to be re-structured - the ideas jump around rather than build logically.

Authors’ response: We thank the reviewer for these comments. We have made major changes to the introduction.

In terms of the goals of this paper (lines 62-64), the first goal is reasonable, but the second could perhaps use a qualifier. This cross-sectional analysis cannot determine temporality or causality, and it seems quite likely that vitamin C deficiency is a marker for many other risk factors (e.g., low SES, low literacy, poor health care access, poor diet quality, etc.) that may impact physical functional health.

Vitamin C measurement preceeded SF-36 (which occurred 18 months from the baseline) and also we controlled for other potential confounders. We acknowledge that there are other confounders such as poor health access and poor diet quality which are not adequately controlled for. We have added an additional sentence in the discussion as below:

“Furthermore, the relationship between vitamin C and self-reported functional health may be biased by residual confounding.”

Materials and Methods:

References should be included for the socioeconomic measures used in this analysis, if available (e.g., the Townsend index).

Authors’ response: We have added the references.

Line 100: Replace “this” with “which”

Authors’ response: We have made the change.

Lines 102-103: What do you mean by “vice versa” - why were there participants who completed the SF-36 but didn’t do the baseline health check? Wasn’t being in the baseline study necessary to be invited to do the follow-up survey?

Authors’ response: SF-36 is a survey sent to all 30,000 who consented to participate at the baseline. Only ~25,000 attended a health check out of 30,000 who consented. Hence the use of vice versa

Line 108: Why only focus on physical functioning and vitality? For example, irritability is a symptom of latent scurvy and would fall under the “role limitation due to emotional problems” domain.

Author’s response: We thank the reviewer for this comment. To prevent a type I error we set out a pirori to examine only the SF-domain we thought would be associated with vitamin C. These were physical functioning and vitality. We think that irritability will be covered in general health and it is a less severe symptom which may not be reliably measured in “role limitation due to emotional problems”.

Line 116: This wouldn’t be a case-control analysis, as this is a cross-sectional study. Cross-sectional studies look at prevalent outcomes (individuals with vitamin C deficiency regardless of when it developed) while a case-control analysis would include only incident cases. You don’t know how long individuals in this population had vitamin C deficiency, so you wouldn’t call this a case-control analysis.

Authors’ response: We have amended this sentence.

“The primary analysis  carried out  identified risk factors for vitamin C deficiency in the UK setting.”

Line 118: These two vitamin C categories (<11.4 vs. ≥11.4 μmol/L) likely include some misclassification around the cutpoint. You could consider looking at low (<11.4) vs.="" optimal="">22.7). Removing the middle group (considered “marginal”) will arguably give you more meaningful comparison groups.

Authors’ response: We thank the reviewer for this important point and have reanalysed table 1 and 2 as advised. Providing a marginal category in table 1. This has made the results of the MVA more robust as we are sure those in the comparison group are unlikely to be misclassified. This was a great suggestion and we were glad to amend the paper. Tables 1 and 2 have been amended and the results section.

Line 118: What do you mean by “assess linearity in categorial variables”?

Author’s response: We have rephrased this sentence.

“Log-likelihood ratio tests were used to assess linearity in continuous variables.”

Line 119: Better to say “Stepwise methods were used to assess which variables…”

Author’s response: We have rephrased this sentence.

Line 122: Don’t need to specify that covariates were added sequentially to the model, as you already explain that you used a stepwise model.

Author’s response: We have rephrased this sentence.

Line 123: replace “there” with “they”

Author’s response: We have rephrased this sentence.

Overall thought about model building process: I’ve used stepwise selection myself, but more and more I’m thinking of it as a part of the process rather than as a way to identify the final model. Giving the computer free rein necessarily means not using your own knowledge to identify which variable make most sense in the model. For example, following is a link that details some concerns about stepwise models and suggests some alternatives: https://towardsdatascience.com/stopping-stepwise-why-stepwise-selection-is-bad-and-what-you-should-use-instead-90818b3f52df

Authors’ response: We absolutely agree about ‘black box’ methods. We used Stata to add each variable sequentially to the model and started with the variables with the lowest level of significance at the univariable level. We examined at each addition the change in the model and improved model fit. We used a combination of both stepwise methods and previous knowledge about likely predictors as well as only considering plausible variables. There are many variables in the EPIC-cohort study but we chose variables that we thought might be plausible predictors using the literature.

Lines 127-129: I understand what you’re saying here, but had to read a few times - try re-wording this for better clarity.

Authors’ response: We have removed the paragraph due to issues of presenting post-hoc power calculations as the sample size was predetermined and power calculations are predominately prospective.  

John M Hoenig & Dennis M Heisey (2001) The Abuse of Power, The American Statistician, 55:1, 19-24, DOI: 10.1198/000313001300339897

Lenth, Russell V. "Post hoc power: tables and commentary." Iowa City: Department of Statistics and Actuarial Science, University of Iowa (2007).

Lines 136-146: Why are you comparing the bottom decile/quartile to all other categories combined? My feeling is that it would be more appropriate to use one of the extreme categories as your reference group, and to show the ORs for all other categories relative to that reference group. This would allow you to look for trends and threshold effects.

Authors’ response: We thank the reviewer for this comment, however, the aim of the study was to assess for the predictors of low levels of Vit C rather than predictors of Vit C itself.

Results:

Line 159: Consider including a third group here - those with optimum vitamin C levels (>22.7)

Author’s response: Thank you for this suggestion we have amended and added a third group according to the literature. As a result of the change, education was included in the final model and binary occupational social class was removed.

Line 165: Referring to a baseline time point here may confuse some readers - this is a cross-sectional analysis that happens to have measured different variables at two different time points. There wasn’t really a follow-up time point, just a second assessment time point. My feeling is that it would be better to simple remove reference to “baseline” here, as it may confuse people about the nature of this study design.

Authors’ response:  We have amended this sentence according to the reviewer’s comment. Please also refer to our explanation regarding vitamin C and SF-36 data.

Out of 30 445 participants who agreed to be part of the study, 22 474 attended the first health check and had a plasma vitamin C sample available in the EPIC-Norfolk study.”

Line 166: It doesn’t seem appropriate to say “after adjustment” here, because you don’t have one key independent variable of interest. You might say “The final regression model included seven risk factors…”

Authors’ response:  We have amended this sentence according to the reviewer’s comment.

“The final regression model included seven risk factors that were statistically associated with vitamin C deficiency”

Lines 166-180: These results are re-stating the information in Table 2. This can be shortened into a brief summary of these results.

Authors’ response: We think that the re-stating the results in table 2 has been completed adequately. We have only included relevant and important association which are relevant to population health. It is also useful for the reader to be able to place our sentence with an OR and CI. Although if it would improve readibility we are happy for all ORs and CI to be removed from the text. We ask the editorial team to advise us on the policy of the journal and whether this needs to be summarised further.

Line 183: Include the word “to” before “those with”.

Authors’ response:  We have amended this sentence according to the reviewer’s comment.

Lines 182-184: Why is this supplemental table necessary? It should be referred to with Table 1 if it is kept, but I’d argue against keeping it as it seems redundant.

Author’s response:  This allows the reader to understand the distribution and we believe some readers would be interested to know the details.

Lines 184-192: Don’t re-state results presented in Table 3 - just summarize

Authors’ response: We think that the re-stating the results in table 3 has been completed adequately. We ask the editorial team to advise us on the policy of the journal and whether this needs summarised further.

Line 198: How much did these domains improve? Don’t need to re-state all these results, but could provide the range of improvement across the different domains.

Authors’ response:  We have amended this sentence according to the reviewer’s comment.

“(0.65-1.68 unit increase in SF-36 per SD increase in vitamin C)”

Table 3: Consider presenting the three categories of vitamin C status (deficiency, borderline, optimum) rather than quartiles; if quartiles are kept, include the ranges for each quartile (i.e., include the range for vitamin C levels within each quartile).

Authors’ response:  We have added the range of each quartile into the results according to the reviewer’s comment.

“The range of vitamin C in each quartile are as follows quartile one:<41.0 µmol/L; quartile two: 41.0 -53.9 µmol/L; quartile three: 54.0 – 65.9 µmol/L; and quartile four: ≥66.0 µmol/L”

Table 3: Model A should be described as being adjusted for age and sex, not “unadjusted + age and sex” (it is adjusted, so calling it unadjusted seems confusing)

Authors’ response:  We have amended this sentence according to the reviewer’s comment.

Basic model + age and sex”

Table 4: Given that all models are adjusted for the same covariates, you could label the first column “independent variable” and include an asterisk there that refers to the covariates included in the models.

Authors’ response:  We thank the reviewer for this comment. We think the table is clear in its current presentation. We defer to the editorial team for their opinion about the style of table 3.

Discussion:

Line 207: Delete “those with”

Authors’ response:  We have amended this sentence according to the reviewer’s comment.

Line 210: Delete the first “who” 

Authors’ response:  We have amended this sentence according to the reviewer’s comment.

Line 215: Say “in contrast to previous studies” then included the citations (although the way you did it is rather fun - it just took me a second to get it)

Authors’ response:  We have amended this sentence according to the reviewer’s comment.

Line 215: What exactly did those previous studies find? A non-linear association with age?

Authors’ response:  We have added the following information to the sentence.

“In contrast to previous studies [8,9,31], this study found older age was linearly associated with vitamin C deficiency, where a quadratic trend was shown in the US study [8], age was not statistically associated with vitamin C [31] and was not examined as a risk factor [9].”

Line 217: Have previous studies looked for this, or have there simply not been any studies of this association? 

Authors' response: We have clarified the sentence.

“Although no previous research has examined this association, this finding is consistent with a wider body of work.”

Line 218: Delete “has”

Authors’ response:  We have amended this sentence according to the reviewer’s comment.

Lines 232-233: How many people in this study would really qualify as excessive drinkers? If 14 units per week refers to having two standard drinks a day, then that’s not excessive for men. It would be interesting to categorize drinking more formally, with separate designations for men and women. For example, the CDC defines heavy drinking as 8 or more drinks per week for women, and 15 or more per week for men. Binge drinking could also be considered (if such data are available).

Authors’ comment: We have categorised alcohol to be in accordance with the new NHS guidelines. We have updated all the analyses in the paper according to this change in the variable categorisation.  

https://www.nhs.uk/live-well/alcohol-support/calculating-alcohol-units/?tabname=alcohol-facts

Lines 236-238 do not seem to fit with the rest of this paragraph

Authors’ response:  We have moved this to another paragraph and reworded.

Line 245: Include a comma after comorbidities

Authors' response: We have amended this sentence.

Lins 246-247: It seems contradictory to say that a finding is both novel and consistent with previous work.

Authors’ response:  We have amended this sentence according to the reviewer’s comment.

Line 247: Definitely too confusing to start a sentence with a parenthetical citation

Authors’ response:  We have amended this sentence according to the reviewer’s comment.

Lines 253-254: Why are all these Authors’s listed here? 

Authors’ response:  Amended in the discussion.

Lines 257-258: Not sure what you mean here. Is “phonotypical” the correct word? The sentence makes sense without that word, but with it I’m not sure what’s going on. :o)

Authors’ response:  We have amended this sentence according to the reviewer’s comment.

Lines 260-270: This paragraph jumps around a bit - would help to re-structure so that the information is presented more logically.

Authors’ response:  We have amended this paragraph according to the reviewer’s comment.

Lines 267-270: One method to address this would be to compare the highs vs. the lows, as I suggested previously.

Authors’ response: We thank the reviewer for this comment, however, the aim of the study was to assess for the predictors of low levels of Vit C rather than predictors of Vit C itself. It would be an interesting research question but not the one that we are addressing in this paper.

Lines 271-275: While you’re re-structuring, include the information about misclassification of vitamin C (lines 267-270) in this paragraph.

Line 277: What does “it” refer to? The SF-36 data?

Authors’ response:  We have amended this paragraph according to the reviewer’s comment.

Line 280: I’m not sure that the “objective” part of the plasma vitamin C measure is really the key strength. You described some important limitations of this measure, but is there any information on how valid these data are likely to be? In line 272 you say “it could be argued that vitamin C levels may fluctuate” - is there any evidence of that? I’m more interested in knowing just how good this biomarker is than in the fact that it’s objective. There are some truly terrible objective biomarkers of nutrition status out there. How good is this one?

Authors’ response:  We have amended this paragraph according to the reviewer’s comment. We argued that it is a good measure of dietary vitamin C. We have removed the word objective as this may be contentious given the comments of other reviewers.

“Other strengths include a large sample size with the ability to control for a myriad of socioeconomic and lifestyle factors, the ability to examine relationships based on vitamin C supplement use and use of plasma vitamin C as a measure of dietary vitamin C.”

Conclusions:

Line 282: This analysis was about vitamin C levels - you didn’t look at whether vitamin C levels were in fact an indicator of fruit and vegetable intake in this population. Which perhaps is fine, as that’s been shown previously, but in this context it seems clearer to simply say “Low plasma vitamin C levels were associated with…”

Authors’ response:  We have amended this paragraph according to the reviewer’s comment.

Overall these conclusions seem reasonable, but I encourage a little more consideration of the fact that the risk factors you identified here are very broad and will relate to many adverse health outcomes. I don’t really feel that you can pinpoint vitamin C deficiency with what you showed here. If a patient comes in complaining of poor health, and the patient is an older male who smokes and drinks and is inactive, it would be odd to say “aha! must be that you have low vitamin C!” - it could just be drinking and smoking and lack of activity.

I feel like the better approach to identifying those at risk for low vitamin C is simply to ask whether people eat fresh fruits and vegetables. If they don't, then worry about their vitamin C.

My biggest concern here is that you frame these results in the context of a public health intervention to identify those at risk for vitamin C deficiency. I don't feel that these results will be useful in that sense (the identified risk factors are not specific enough), but I do feel that this would be a strong paper if the emphasis was shifted to a discussion of how vitamin C deficiency is associated with poor functional health in general. Even if people don't have scurvy, they are going to have general poor health if they're micronutrient deficient. 

Authors’ response: We have amended the conclusion according to the reviewer's comments.

“Low plasma vitamin C levels were associated with poor physical functioning, an established risk factor of objective health outcomes as well as a valid quality of life outcome on its own right. We have also identified risk factors for vitamin C deficiency. Vitamin C deficiency was associated with poor self-reported functional health, which may reflect that this population have symptoms of vitamin C deficiency. Whilst scurvy is rare, populations with poor general health are likely to be micronutrient deficit and may benefit from increased fruit and vegetable intake.”

Round  2

Reviewer 2 Report

I am satisfied with the authors' responses to points raised. Please proofread the paper well before publishing.

Author Response

We thank the reviewer for their comments. We have made the necessary adjustments to the manuscript. 

Reviewer 3 Report

I feel that the revisions have improved this manuscript in terms of content, but there are areas where writing quality seems to have suffered a bit. For example, the introduction doesn't follow a logical outline, and the conclusion jumped out at me as an example of a poorly-constructed paragraph; the second sentence seems out of place, and the meaning of the third sentence isn’t clear. These are just a couple examples, but my overall impression here is that editing for clarity is needed throughout the manuscript.

I feel that this manuscript would benefit from a round of editing by the authors themselves, rather than moving on directly to the editorial staff. The authors know best what they're trying to convey, and with a fresh read through will be able to identify many of the issues with sentence construction and clarity that are present here.

I feel that the content is solid at this point, but is not communicated as clearly as could be desired. 

Author Response

We thank the reviewer for their thorough comments. We have made some minor adjustments to the manuscript.